# Population Pharmacokinetic Model of Adalimumab Based on Prior Information Using Real World Data

**DOI:** 10.3390/biomedicines11102822

**Published:** 2023-10-18

**Authors:** Silvia Marquez-Megias, Ricardo Nalda-Molina, Patricio Más-Serrano, Amelia Ramon-Lopez

**Affiliations:** 1School of Pharmacy, Miguel Hernández University, 03550 San Juan de Alicante, Spain; silvia.marquez@goumh.umh.es (S.M.-M.); mas_pat@gva.es (P.M.-S.); aramon@umh.es (A.R.-L.); 2Alicante Institute for Health and Biomedical Research (ISABIAL-FISABIO Foundation), 03010 Alicante, Spain; 3Clinical Pharmacokinetics Unit, Pharmacy Department, Alicante University General Hospital, 03010 Alicante, Spain

**Keywords:** pharmacokinetics, drug monitoring, adalimumab, monoclonal antibodies, inflammatory bowel diseases, Crohn’s disease, ulcerative colitis

## Abstract

Adalimumab is a fully human monoclonal antibody used for the treatment of inflammatory bowel disease (IBD). Due to its considerably variable pharmacokinetics and the risk of developing antibodies against adalimumab, it is highly recommended to use a model-informed precision dosing approach. The aim of this study is to develop a population pharmacokinetic (PopPK) model of adalimumab for patients with IBD based on a literature model (reference model) to be used in the clinical setting. A retrospective observational study with 54 IBD patients was used to develop two different PopPK models based on the reference model. One of the developed models estimated the pharmacokinetic population parameters (estimated model), and the other model incorporated informative priors (prior model). The models were evaluated with bias and imprecision. Clinical impact was also assessed, evaluating the differences in dose interventions. The developed models included the albumin as a continuous covariate on apparent clearance. The prior model was superior to the estimated model in terms of bias, imprecision and clinical impact on the target population. In conclusion, the prior model adequately characterized adalimumab PK in the studied population and was better than the reference model in terms of predictive performance and clinical impact.

## 1. Introduction

Adalimumab is a fully human recombinant immunoglobulin G (IgG) monoclonal antibody that inhibits the binding of tumor necrosis factor (TNF) to its receptors, decreasing the process of inflammation. Adalimumab is increasingly used for the treatment of moderate-to-severe inflammatory bowel disease (IBD) patients both in induction and maintenance phases that had an inadequate response to corticosteroids, immunomodulators or other biologic therapies [1,2].

Numerous studies have demonstrated the association between higher serum drug levels of adalimumab and better clinical outcomes [3,4]. The exposure target depends on whether patients are diagnosed with Crohn’s disease or ulcerative colitis and on the desired therapeutic objective, such as clinical, endoscopic, biochemical or histologic remission, although the most accepted target is the endoscopic remission [5]. In relation to this, some studies indicated that 8–12 mg/L trough serum concentrations (TSC) of adalimumab are required to achieve mucosal healing and endoscopic remission in 80–90% of IBD patients [5,6]. In fact, several studies have shown that after long periods of subtherapeutic drug levels, approximately 40% of patients with IBD can experience an irreversible disease worsening or develop antibodies against adalimumab (AAA) and, therefore, require dose escalation or even a switch to another drug [7,8,9,10,11,12,13,14]. A prospective study evidenced that having an improvement in clinical outcomes from dose escalation is difficult to achieve once they experience a loss of response [15]. Consequently, a therapeutic range of 8–12 mg/L has been considered as the therapeutic target in the clinical setting.

Model-Informed Precision Dosing (MIPD) is a Bayesian approach based on the use of population pharmacokinetic (PopPK) models to calculate the individual pharmacokinetic (PK) parameters for each patient. These individual PK parameters are used to achieve the optimal dose regimen to balance efficacy and toxicity and improve the treatment outcomes for each individual patient [16]. A multicenter retrospective study in patients treated with adalimumab indicated that the MIPD approach can prevent immunogenicity, lowering the risk of developing AAA and achieving better long-term outcomes in terms of IBD-related surgery or hospitalization. Moreover, it also proved to be more cost-effective compared to empirical and/or reactive dose optimization [17].

However, there are six PopPK models for adalimumab and IBD patients published in the literature. All models had a similar structure (one-compartment model), although the included covariates and the values of the PopPK parameters differ among them [1,18,19,20,21,22]. Even though a PopPK model implemented from the literature can suit a population in the clinical setting, it is convenient to adapt this PopPK model to the studied population, to re-estimate the parameters and to evaluate the inclusion of potential new covariates to obtain more accurate results in the dose optimization. In a previous study, the predictive performance of these PopPK models was externally evaluated in the clinical setting [23]. This study, conducted by our research group, concluded that the PopPK model developed by Ternant et al. (reference model) was better than the others in terms of model adequacy and predictive performance [18]. However, the EBEs of the individual CL/F were found to be biased when compared with the mean population values in the models.

Therefore, the aim of this study is to optimize a PopPK model of adalimumab for IBD, previously selected from the literature, considering its improvement in predictive performance and clinical impact, with the subsequent application in the clinical setting for MIPD.

## 2. Materials and Methods

### 2.1. Study Design

A retrospective observational study was conducted at the Dr. Balmis General University Hospital of Alicante on patients with IBD in treatment with adalimumab who followed an MIPD program between 2014 and 2022.

### 2.2. Patients and Data Collection

This study included patients with IBD who underwent adalimumab treatment at the Dr. Balmis General University Hospital of Alicante, Spain. Participants with at least two adalimumab TSC were eligible for inclusion. Patients treated with monoclonal antibodies other than adalimumab, such as infliximab, vedolizumab or ustekinumab, and subjects who were diagnosed with autoimmune diseases other than IBD, such as rheumatoid arthritis, psoriasis or ankylosing spondylitis, were excluded from this study.

The covariates evaluated in this study included age, sex, height, body weight, body mass index, IBD type, serum albumin, serum C-reactive protein, fecal calprotectin, AAA status and AAA serum concentration, use of concomitant immunomodulators, previous anti-TNF treatment and whether adalimumab originator or biosimilar was used. For missing covariates, the mean value of this covariate for a given patient was imputed. If any patient had no available value of a covariate, the mean value of that covariate of the rest of the patients was imputed.

TSC and AAA were determined using an enzyme-linked immunosorbent assay LISA TRACKER Duo Drug + ADAb (TheraDiag^®^, Paris, France). The limits of quantification for SC and AAA were 0.1 mg/L (range 0.1–16 mg/L) and 10 ng/mL (range 10–2000 ng/mL), respectively. Patients were considered as positive for AAA if titers were above 10 ng/mL on at least one occasion.

### 2.3. Model Development and Evaluation

The reference model was the one selected among all available models in the literature, according to a predictive performance evaluation published elsewhere [18,23]. Briefly, the model, developed with Monolix 4.3.2, comprises a one-compartment model with first-order absorption and linear elimination and was parameterized in terms of apparent clearance (CL/F), apparent volume of distribution (V/F) and absorption constant (ka) with a combined residual error model. The presence of AAA was included as a categorical covariate on CL/F.

Initially, the reference model was refitted by estimating the PK population parameters using all the available TSC of patients in Monolix software V.2023R1 [24]. The model structure was the same as the reference model, including the covariate model. Ka and the effect of AAA on CL/F were fixed to the published value.

The use of informative priors in the model was also considered by using the option of maximum a posteriori estimation in Monolix. The estimated values and the relative standard error (RSE) of the estimation of the parameters of the reference model were used to define the prior. To evaluate the appropriateness of the prior for each parameter, priors were set individually using an informative prior, whereas the rest of the parameters were kept as noninformative. The informative priors that reduce the RSE of the parameter estimations and result in better predictive performance would be retained in the model.

### 2.4. Covariate Analysis

Covariate analysis was based on physiological plausibility and visual graphical inspection of the relationships between Empirical Bayes Estimates (EBEs) of the PK parameters and the covariates. Statistical significance (*p* < 0.01) was further evaluated individually in the PK model using a stepwise forward addition and backward elimination covariate model-building methodology.

### 2.5. Model Selection

The improvement in predictive performance was the criterion for model selection. A decrease in the RSE of the parameter estimation was also considered for the inclusion of informative priors.

To evaluate predictive performance, the individual predictions of the last TSC were estimated for each patient, using EBEs of the individual PK parameters. These last TSCs, named the “last observed TSC”, were left out and not used to calculate the EBEs of the individual PK parameters. Bias and imprecision were then calculated using the last observed TSC by comparing them with their individual predictions.

The mean prediction error (MPE, Equation (1)) and root mean square prediction error (RMSPE, Equation (2)) were calculated for bias and imprecision, respectively.
(1)$$MPE=\frac{\sum \left(\hat{Y}-Y\right)}{n}$$
(2)$$RMSPE=\sqrt{\frac{\sum {\left(\hat{Y}-Y\right)}^{2}}{n}}$$

In both equations, Y-hat represents the individual-predicted adalimumab concentration, Y represents the observed adalimumab concentration, and n is the number of observations.

Additionally, a Predicted-Corrected Visual Predictive Check (pcVPC) for the reference and the final model was performed to evaluate predictive performance. Graphical evaluation, e.g., residual vs. predicted, observed vs. predicted and NPDE, was also evaluated.

A bootstrap of the data was performed to compare statistical significance of the differences between bias and imprecision of the different models.

### 2.6. Model Validation

A numerical Predictive Check (NPC) was performed as an internal validation of the model adequacy of each model. NPC quantitatively compares the cumulative observed adalimumab concentrations that correspond to the model-simulated percentiles with their expected concentrations that represent the 50th percentile of the observed concentrations, as well as the 95% confidence interval (CI) for the 50th percentile of the predicted concentrations.

The accuracy and robustness of parameter estimates were evaluated using a bootstrap with 500 replicates constructed by sampling individuals with replacements from the original dataset. Model parameters were estimated for each bootstrap replicate and were used to estimate the mean and 95% CI from the individual replicates.

These databases generated with the bootstrap were also used to validate predictive performance by calculating the mean and 95% CI of bias and imprecision of each model for each of the 500 replicates.

### 2.7. Clinical Impact

The evaluation of the clinical impact of PopPK models was performed by calculating the true positives and false positives of the predictions of the last TSC for each model compared to the last observed TSC. It is worthwhile to mention again that the last TSCs were left out to calculate the EBEs of the PK parameters for each model. Three different scenarios were considered to calculate true and false positives, assuming three concentration ranges: below the target; within the target; and above the target. The last observed TSC was considered the standard reference for each concentration range. True and false positives were calculated by comparing the coincidences and discrepancies with the predicted TSC with each PopPK model, corresponding to such last observed TSC. The target interval of the TSC that was considered in this study was within 8–12 mg/L for clinical response or remission [6,15].

The 95% CI of true and false positives in each scenario for each model was calculated with the bootstrap.

### 2.8. Software

The software used for model development was Monolix 2023R1^®^ [24]. The statistical analysis, data visualization and validation were performed using R software v4.2.2 [25], implemented in RStudio 2022.07.2 + 576 [26].

## 3. Results

### 3.1. Patient Characteristics

The resulting dataset comprised 54 IBD patients in treatment with adalimumab with at least two TSCs. Approximately 85% of the patients were diagnosed with Crohn’s disease and 15% with ulcerative colitis. The summary of the characteristics of the studied population compared to the population of the reference model is listed in Table 1.

As an induction phase, 43 patients were treated subcutaneously with 160/80 mg and 2 patients with 80/40 mg at weeks 0/2. The information regarding the induction phase of the other nine patients was not available in their medical histories. Following this phase, all patients were treated with 40 mg of adalimumab every other week. A total of 148 TSC, 19 of them in the induction phase, were available for analysis. 68.2% of TSC were below 8 mg/L, 16.2% between 8 and 12 mg/L and 15.6% over 12 mg/L. AAA were detected in nine patients (17%). 22 patients were on a concomitant immunomodulator (6-mercaptopurine, aminosalicylate, azathioprine, corticosteroids, methotrexate or combined). 39 patients were treated with adalimumab originator (HUMIRA^®^), and 15 patients were treated with biosimilars (10 patients with HYRIMOZ^®^ and 5 patients with IDACIO^®^).

### 3.2. Model Development, Covariate Analysis and Evaluation

Due to the lack of serum concentrations in the absorption phase in the dataset and the small number of AAA-positive patients, ka and the covariate of AAA on CL/F were fixed to the values of the reference model, 0.00625 1/h and 4.5, respectively.

In the first step, all the parameters were estimated, keeping the model structure of the reference model.

Figure 1 shows the relationship between EBEs of CL/F and albumin with a statistically significant slope (*p* < 0.001). In the forward inclusion step of the covariate modeling, only albumin was found to be a significant covariate influencing CL/F, with an improvement in the Objective Function Value (OFV) of 12 (*p* < 0.001).
(3)$$CL/F={CL}_{pop}\cdot (1+AAA\cdot {cov}_{AAA-CL/F})\cdot {\left(\frac{ALB}{mALB}\right)}^{{cov}_{ALB-CL/F}}$$

CL/F is defined according to Equation (3), where AAA is a categorical covariate representing the absence and presence of AAA, and albumin is a continuous covariate weighted to the mean value (3.77 g/dL) in the studied population (mALB). In addition, the inclusion of albumin as a covariate on CL/F resulted in better performance in terms of bias and imprecision. Model structure and code have been added as a Appendix A.

In the second step, the use of priors in different parameters was evaluated. The inclusion of informative priors in the IIV of CL/F and the IIV of V/F resulted in a substantial reduction in RSE, not only on these parameters but also in the parameters estimated without priors. The resulting RSE using priors decreased from 30.6% to 3.4% for the IIV of CL/F and from 114.5% to 1.4% for the IIV of V/F, compared to the model where all parameters were estimated. For the remaining parameters, the inclusion of priors did not improve the fit, neither in terms of RSE nor predictive performance. Additionally, residual unexplained variability was modeled using a proportional error model due to the high RSE of the additive error (83.8%). This model would be considered the final model. 

The final model shows a considerable reduction in bias compared to the reference model and a similar dispersion of Individual Residuals (IRES), as is shown in Figure 2. Table 2 shows bias and imprecision for the reference and the final model and the differences between them. The final model behaves better in terms of bias and imprecision. The 95% CI of the differences, calculated with the bootstrap, shows statistical differences in bias but not in imprecision.

The pcVPC for the reference and the final model, represented in Figure 3, shows that the final model performs better than the reference model. The same results are observed in Observed vs. Predicted (Appendix A) and NPDE (Appendix A) plots, available in the Appendix A. 

The values of each parameter of the final model compared to the reference model are listed in Table 3.

### 3.3. Model Validation

The NPC of the reference and the final model are represented in Figure 3. The final model shows a better performance compared to the reference model.

### 3.4. Clinical Impact

Figure 4 shows true and false positives of the individual predictions of the last observed TSC of the final model for each scenario. Among all the last observed TSCs in the dataset, 36 TSCs fell below target, 8 TSCs fell within the target, and 10 TSCs fell above target. Table 4 shows true and false positives of the predictions of the last TSC and the differences between the reference and the final model for each scenario. In all cases, the final model performs better than the reference model in terms of true positives and false positives. 

## 4. Discussion

The MIPD approach can be a useful tool to optimize the dose of drugs with high pharmacokinetic variability. To apply this methodology in the clinical routine, it is common to use PopPK models found in the literature due to the difficulties of building in-house PopPK models with the available data in hospitals.

The model reference was based on 341 adalimumab serum concentrations derived from 65 patients during a follow-up of 500 days, although only Crohn’s disease patients were included in this study. Regarding the analytical assay, ELISA and Double-antigen ELISA were used to measure adalimumab TSCs and AAA, respectively. However, it is not specified which limit of titers was used to consider the patients as AAA positives. The value of this limit is crucial in the estimation of the proportion of positives and, therefore, its quantitative effect on CL/F. Moreover, biochemical covariates such as albumin, C-reactive protein or fecal calprotectin were not available.

The inclusion of AAA and albumin in the final model as covariates of CL/F was found to statistically improve the OFV and also reduce the interindividual variability in CL/F. The association between the presence of AAA and the increase in adalimumab CL/F, leading to lower adalimumab concentrations, has been reported in numerous studies [8,9,10,11,12,13]. In our study, the presence of AAA was found to be a determinant covariate. However, the estimation of the effect of AAA on CL/F was not possible due to the small number of patients’ positives for AAA and, therefore, it was fixed to the reference model value.

The results of this study suggest that patients with lower albumin have a higher CL/F. In addition, CL/F increases 12-fold as albumin rises from the lowest value (1.97 g/dL) to the highest value (4.96 g/dL). Therefore, patients with lower albumin require higher doses to reach the desired target; otherwise, plasma concentration would fall into the infratherapeutic range. Several studies demonstrated the correlation of higher albumin levels with higher response rates to infliximab and adalimumab [27,28,29,30,31,32,33,34]. In fact, albumin was a significant covariate on CL in a considerable number of previously published PK models of infliximab for IBD [35]. In contrast, other studies that developed PopPK models of adalimumab in Crohn’s disease [19,21] or IBD patients [22] observed that higher albumin levels were associated with lower adalimumab CL/F and higher serum levels, considering albumin as an influential inflammatory marker of adalimumab clearance, although, finally, they did not include it as a covariate in the PopPK model. However, albumin is also a well-known surrogate marker of disease that could exacerbate with an increase in CL. Therefore, further studies are necessary to establish whether albumin has a direct impact on CL or the change in CL and, consequently, the change in plasma concentration of Adalimumab has an impact on the albumin.

Several studies have shown that fecal calprotectin and C-reactive protein are reliable markers of endoscopic activity and therapeutic response in IBD patients [36,37,38]. In fact, C-reactive protein and fecal calprotectin showed a positive influence on adalimumab CL/F in a PopPK model of adalimumab developed for IBD that included the latter as a continuous covariate [22]. However, the association of TSC and C-reactive protein or fecal calprotectin was not found in our data.

Body weight was included as a covariate on CL/F in four PopPK models of adalimumab in Crohn’s disease [1,19,21] or IBD patients [22] and on V/F in one of them [19]. However, body weight, lean body weight and body mass index did not show a significant relationship with any PK parameter of adalimumab in our population.

A priori information could be used to stabilize the estimation of the model parameters when the data available are sparse. Several studies showed that the use of priors allowed a better fit to the new data than fixing the parameters [39,40,41,42]. Moreover, the model built with priors in our study was more stable, provided a better fit of the data and reduced IIV. In this line, other authors also obtained similar results [43].

In order to mimic the real-world conditions, predictive performance was calculated with TSCs that were left out for the calculation of EBEs of the PK parameters. The results of predictive performance in terms of bias and imprecision were −1.79 and 4.14 for the reference model and −0.849 and 3.99 for the final model, respectively. The bootstrap analysis of predictive performance showed statistically significant differences in terms of bias.

Regarding the clinical impact, the final model obtained 15% more true positives (39 vs. 33) than the reference model. Similarly, the final model obtained 30% less false positives than the final model. Therefore, the final model better predicts the need for dose modification.

One of the main limitations of this study is its retrospective design, where patients were selected for MIPD based on the clinical decision of the physician, which implies a potential bias related to the disease severity. This potential bias could lead to an underestimation of the mean values and variance of albumin, C-reactive protein and fecal calprotectin in the IBD population. Another limitation inherent to the clinical setting is that only TSCs were available since data were obtained from the clinical setting; therefore, there is a lack of serum concentrations in the absorption phase and, consequently, ka could not be estimated, so it was fixed to the value of the reference model.

In conclusion, the developed PopPK model, using informative priors in IIV of CL/F and IIV of V/F based on the reference model, adequately characterized adalimumab PK in the studied population and performed better than the reference model in terms of predictive performance. The main structural difference between both models was the inclusion of albumin as a meaningful covariate on CL/F. To our knowledge, this is the first PopPK model of adalimumab in IBD that identified albumin as a covariate on CL/F. Additionally, the final model significantly improves the clinical impact on the target population and could allow a more accurate dose optimization and an improvement of adalimumab treatment efficacy.

## Figures and Tables

**Figure 1 biomedicines-11-02822-f001:**
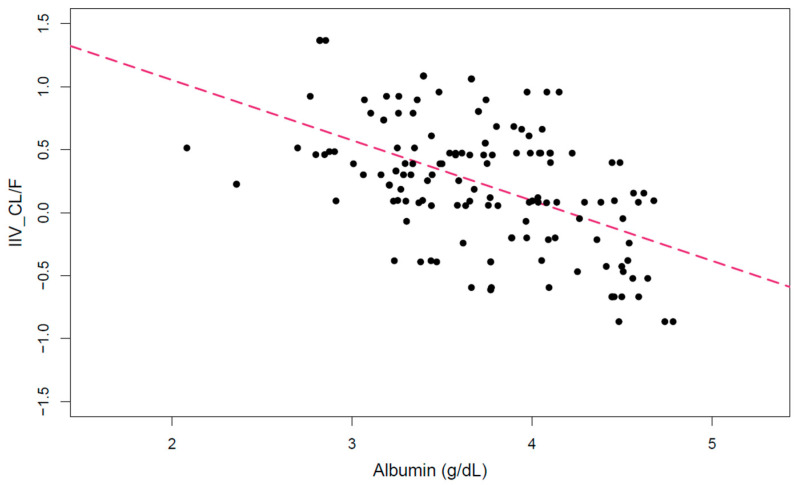
Interindividual variability of apparent clearance versus albumin.

**Figure 2 biomedicines-11-02822-f002:**
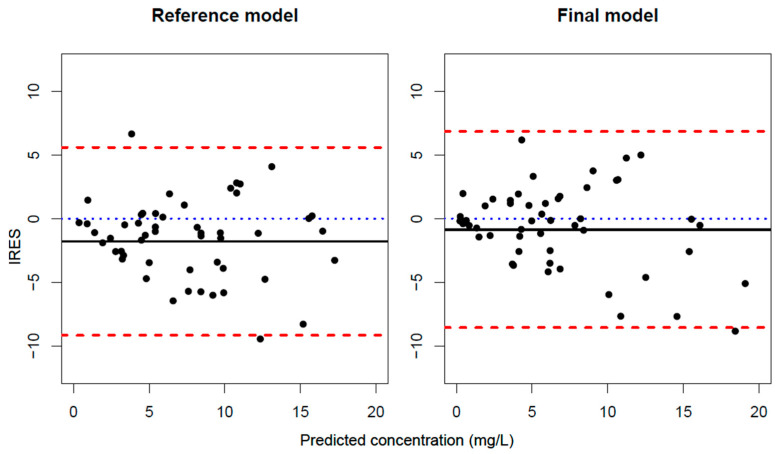
Individual residual (IRES) versus the individual predicted of the last observed trough serum concentrations (TSC) for the reference and the final model. The mean IRES (black solid line) represents the bias of each model; red dashed line represents the 5th and 95th percentile for IRES; blue dotted line the line corresponding to 0.

**Figure 3 biomedicines-11-02822-f003:**
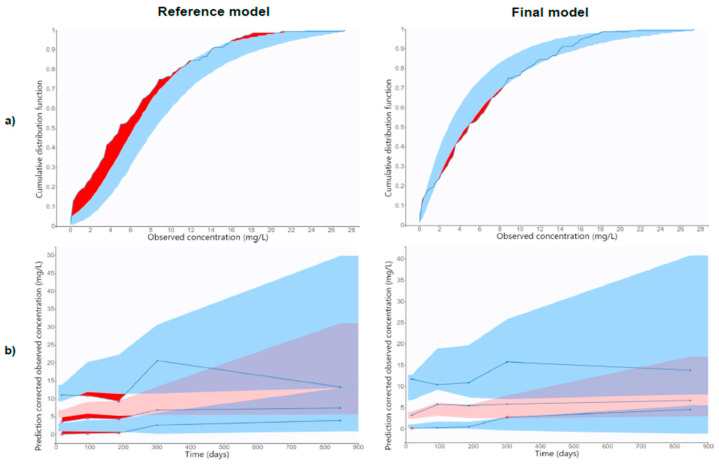
(**a**) NPC of the reference and the final model. Blue solid line depicts the empirical distribution. Blue shaded area represents the 95% confidence interval for the median of the predictions, and the red shaded areas represent the outliers. (**b**) pcVPC of the reference and the final model. Blue solid lines represent the 5th, 50th and 95th percentiles of the observed concentrations; Blue shaded areas represent the 95% confidence interval of the 5th and 95th percentiles of the predictions; pink shaded area represents the 95% confidence interval for the 50th percentile of the predictions, and red shaded areas represent the outliers. The RSE of the estimated PK parameters in the final model was below 50% in the bootstrap analysis. No significant differences were observed between the mean values of the PK parameters in the bootstrap analysis of the final model. Moreover, estimated PK parameters were within the 95% CI of the parameters obtained in the bootstrap.

**Figure 4 biomedicines-11-02822-f004:**
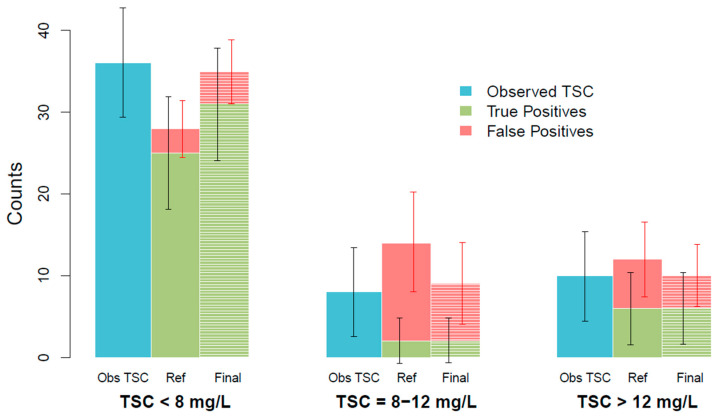
Clinical impact of the reference (Ref) and the final model (Final) predictions compared to the last observed trough serum concentrations (Obs TSC) in the different scenarios. Black arrows represent the 95% CI of the last observed TSC and the true positives of the last TSC predictions, and red arrows represent the 95% CI of the false positives of the last TSC predictions.

**Table 1 biomedicines-11-02822-t001:** Summary of the characteristics of the included patients in the reference and the final model.

Characteristics	Population of the Reference Model	Population of the Final Model
Patients	65	54
Age (yr) †	37 (17–61)	43.5 (11–89)
Sex, male, n (%)	16 (25%)	30 (55.6%)
Weight (kg) †	68 (43–109)	66.5 (34.8–94.0)
Body mass index (kg/m^2^) †	NA	22.84 (14.1–32.03)
Albumin (g/dL) †	NA	3.86 (1.97–4.96)
Prealbumin (mg/dL) †	NA	24.2 (9.0–37.0)
CRP (mg/L) †	NA	0.770 (0.0575–6.680)
FCP (mg/kg) †	NA	513 (25–3600)
IBD type, CD, n (%)	65 (100%)	46 (85.2%)
Adalimumab originator (Humira^®^), n (%)	NA	38 (70.4%)
Prior treatment with infliximab	NA	35 (64.8%)
Concomitant immunomodulator, n (%)	NA	22 (40.7%)
6-Mercaptopurine	NA	1 (4.6%)
Aminosalicylate	NA	3 (13.6%)
Azathioprine	NA	6 (27.3%)
Corticosteroids	NA	5 (22.7%)
Methotrexate	NA	2 (9.1%)
Combined	NA	5 (22.7%)
Adalimumab serum samples	341	148
Adalimumab serum concentrations (mg/L) †	NA	4.90 (0.10–27.4)
AAA positive, n (%)	9 (13.8%)	9 (16.7%)
AAA serum concentrations (mg/L) †,‡	NA	115 (15–459)

NA = not available; CRP: C-reactive protein; FCP: fecal calprotectin; IBD: inflammatory bowel disease; CD: Crohn’s disease; AAA: antibodies against adalimumab. † Median and range of population used to develop the reference model and the final model. ‡ Median and range of patients with presence of antibodies against adalimumab.

**Table 2 biomedicines-11-02822-t002:** Bias and imprecision with the 95% confidence interval for the reference and the final model.

	Models		Bootstrap Results (n = 500)	
Model	Bias (95% CI)	Imprecision (95% CI)	Bias (95% CI)	Imprecision (95% CI)
Reference Model	−1.79 (−2.82 : −0.793)	4.14 (3.11 : 5.09)	−1.78 (−2.76 : −0.804)	4.10 (3.12 : 5.09)
Final Model	−0.849 (−1.86 : 0.160)	3.99 (2.43 : 5.33)	−0.854 (−1.87 : 0.160)	3.90 (2.52 : 5.28)
Difference	−0.939	0.150	0.927 (0.353 : 1.46)	0.200 (−0.670 : 1.08)

CI: Confidence Interval.

**Table 3 biomedicines-11-02822-t003:** Population pharmacokinetic parameters of the reference model and the final model.

	Reference Model	Final Model		Bootstrap Results (n = 500)	
	Estimate (%RSE)	Estimate (%RSE)	95% CI	Mean Value (%RSE)	95% CI
CL/F (L/h)	0.0175 (9%)	0.0312 (10.9%)	0.0246 : 0.0378	0.0314 (12.4%)	0.0234 : 0.0391
ALB_CL/F	-	−2.33 (2.8%)	−2.46 : −2.21	−2.36 (43.8%)	−4.39 : −0.335
V/F (L)	13.5 (10%)	7.76 (24.1%)	4.09 : 11.42	7.70 (19.9%)	4.69 : 10.7
IIV_CL/F	0.65 (10%)	0.667 (15.5%)	0.464 : 0.869	0.666 (3.4%)	0.623 : 0.710
IIV_V/F	0.48 (19%)	0.477 (33.9%)	0.160 : 0.794	0.474 (1.4%)	0.460 : 0.487
Proportional error	0.15 (16%)	0.547 (8.4%)	0.458 : 0.637	0.543 (8.7%)	0.451 : 0.636
Additive error (mg/L)	1.8 (8%)	-	-	-	-

%RSE, relative standard error; CI: confidence interval; CL/F: apparent clearance; V/F: apparent volume; ALB: albumin; IIV: interindividual variability.

**Table 4 biomedicines-11-02822-t004:** True and false positives of the predictions of the last TSC and the differences between the reference and the final model for each scenario.

	TSC < 8 mg/L		TSC = 8–12 mg/L		TSC > 12 mg/L	
	True Positives	False Positives	True Positives	False Positives	True Positives	False Positives
Reference model	25.0 (18.1 : 31.9)	2.95 (−0.50 : 6.41)	2.04 (−0.71 : 4.80)	12.1 (5.95 : 18.1)	5.93 (1.53 : 10.3)	6.04 (1.47 : 10.6)
Final model	30.9 (24.0 : 37.8)	3.99 (0.0720 : 7.90)	2.07 (−0.676 : 4.81)	7.00 (1.99 : 12.0)	6.01 (1.62 : 10.4)	3.99 (0.189 : 7.80)
Difference	5.90 (1.50 : 10.4)	1.04 (−0.836 : 2.90)	0.0300 (−2.80 : 2.85)	−5.05 (−10.0 : −0.102)	−0.0800 (−2.71 : 2.87)	−2.04 (−4.74 : 0.660)

TSC: trough serum concentrations.

## Data Availability

The datasets used and/or analyzed during the current study are available from the corresponding author on reasonable request.

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
