# Peer review of "Population Pharmacokinetic Model of Adalimumab Based on Prior Information Using Real World Data"

_biomedicines, 2023, doi:10.3390/biomedicines11102822_

Round 1

Reviewer 1 Report

The article “Population pharmacokinetic model of adalimumab based on prior information using real world data” addresses an important topic. I have following comments/suggestions,

1. Model development and evaluation: kindly make it clear here what platform was used to develop the reference model?

2. Covariate analysis: Did the author used the reported covariates in the previous studies?

3. The authors are advised to provide the model structure and code as supplementary information. 

Reviewer 2 Report

The topic fits well into this special issue in my assessment. 

I have read this paper with great interest, and with a background on clinical pharmacology, commonly involved in popPK studies, but not being myself a 'hardcore modeller'. 

The findings (albumin as disease severity indicator, the presence of AAA) have an additional impact of CL/F is very reasonable anyhow, but the nice part is the link with the 'impact' analysis. 

just some minor suggestions, in my assessment to further improve the paper

IBD type, what do you with this (line 39)

I understood that the currently accepted target is endoscopic remission. If so, i would further stress this (line 39)

with subtherapeutic level, you refer to the lower threshold of the therapeutic range, so 8 mg/L ?

I have some concerns on the assays used, as ELISA tests are notorious for in between assay variability. Is this a limitation to be added, or are the test CE labelled and validated to a standard ? 

I would suggest to make it somewhat clearer that albumin is likely a disease marker. 

Looking to figure 2, there still seems to be a minor trend with higher concentrations ? 

Figure 4, i undestand that the counts are absolute values, and that's ok, but i would suggest to add the total number to the legend of the figure. (and check if the colors look still sufficiently different if printed black-white, otherwise it is perhaps bettter to adapt ? 

Reviewer 3 Report

There are major methodological issues with the work that prevents from publication in my view. 

1. there are already numerous popPK models in the target population for ADA. Unfortunately the current work is based on no measured concentration in the absorption phase, which can then by definition may not be modelled. Therefore, the authors fixed the absorption paramateres, but then as a result the whole work is mostly just a repetition of modelling excercise originating in the refrence model (previously published). 

2. the authosr claim to improve the previously published model, which is wrong. As the measured data seem to be only trough values, and the exploration only targets on the elimination phase, while the absorption is ignored (or rely on refrence model) the outcome is severely biased and overrates the elimination (as well as distribution) factors. 

3. the final model performance is suboptimal anyway as apparent from the NPC. The authors should also demonstrate the model validity by providing standard observed vs predicted concentration plot, visual predictive check (VPC), normalized prediction distributed errors (NPDE) vs time and pop. predisctions. 

Round 2

Reviewer 3 Report

Unfortunately, my concerns are not solvable by adding one short sentence on line 366 "therefore, there is  not information in the absorption phase." This cannot be considered as appropriate description of major methodological limitation of thudy, which is that the model just repeats the absorption characteristics from previously published data and the overall final model over rates elimination. 
